# Physically-Based Face Rendering
# for NIR-VIS Face Recognition

**Yunqi Miao***
Warwick Manufacturing Group
University of Warwick
Yunqi.Miao.1@warwick.ac.uk

**Alexandros Lattas***
Imperial College London
and Huawei
a.lattas@imperial.ac.uk

**Jiankang Deng**[†]
Huawei
and InsightFace
j.deng16@imperial.ac.uk

**Jungong Han**
Department of Computer Science
Aberystwyth University
jungonghan77@gmail.com

**Stefanos Zafeiriou**
Department of Computing
Imperial College London
s.zafeiriou@imperial.ac.uk

## Abstract

Near infrared (NIR) to Visible (VIS) face matching is challenging due to the significant domain gaps as well as a lack of sufficient data for cross-modality model training. To overcome this problem, we propose a novel method for paired NIR-VIS facial image generation. Specifically, we reconstruct 3D face shape and reflectance from a large 2D facial dataset and introduce a novel method of transforming the VIS reflectance to NIR reflectance. We then use a physically-based renderer to generate a vast, high-resolution and photorealistic dataset consisting of various poses and identities in the NIR and VIS spectra. Moreover, to facilitate the identity feature learning, we propose an IDentity-based Maximum Mean Discrepancy (ID-MMD) loss, which not only reduces the modality gap between NIR and VIS images at the domain level but encourages the network to focus on the identity features instead of facial details, such as poses and accessories. Extensive experiments conducted on four challenging NIR-VIS face recognition benchmarks demonstrate that the proposed method can achieve comparable performance with the state-of-the-art (SOTA) methods without requiring any existing NIR-VIS face recognition datasets. With slightly fine-tuning on the target NIR-VIS face recognition datasets, our method can significantly surpass the SOTA performance. Code and pretrained models are released under the insightface[2] GitHub.

## 1 Introduction

To overcome the problem that the conventional VISible (VIS) images face recognition methods generally fail to achieve a satisfactory performance under poor illumination, face recognition across Near InfraRed (NIR) images and VIS images has recently gained increasing attention in the computer vision community [21, 22, 12, 13]. However, due to the lack of sufficient NIR-VIS data, the training of the NIR-VIS face recognition network is prone to be over-fitting [22].

To alleviate overfitting, previous works attempt to generate large-scale NIR-VIS face images by transferring the VIS images to NIR ones [31, 40, 54, 52]. However, the image-to-image translation based methods are limited by the number of data in the source domain and the diversity of generated images [12]. Recently, unconditional generative models are employed to synthesize heterogeneous

---

*Equal contribution. †Corresponding author. This work is done when Yunqi Miao is an intern at Huawei.
[2]https://github.com/deepinsight/insightface/tree/master/recognition

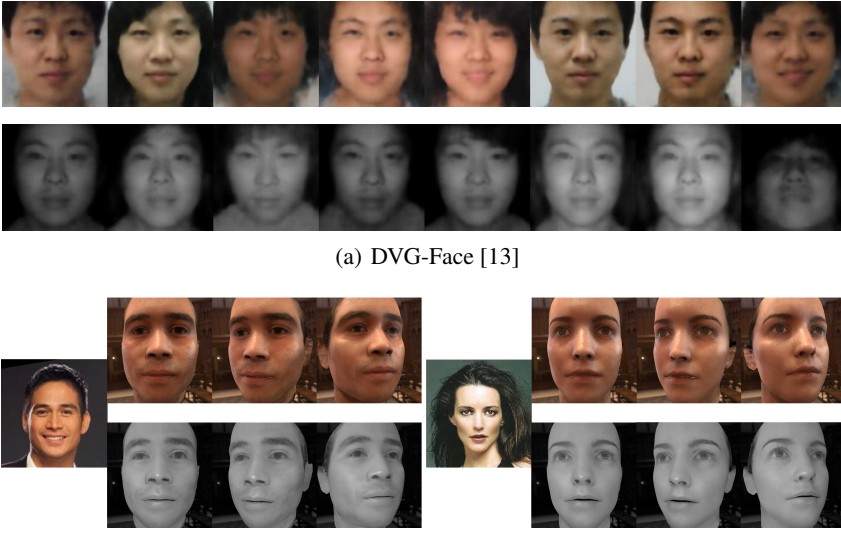

(a) DVG-Face [13]

(b) Ours

Figure 1: (a) Multiple VIS (top row) and NIR (bottom row) face images of the same identity generated by DVG-Face [13], and (b) our method, using two subjects from the CelebA dataset [35].

face image pairs from noise [12, 13] and achieve state-of-the-art performance by adopting the various intra-changes, such as poses and illumination, of the target NIR-VIS datasets during the generation. Although the intra-class diversity is considered, only one NIR-VIS pair is generated for each identity in [13], which limits the potential of synthesized face images in the NIR-VIS face recognition task. When generating multiple NIR-VIS image pairs for a given identity by using [13], we observe that the identity consistency cannot be well preserved, as shown in Fig. 1(a). Additionally, the appearance variations of generated images are based on the target NIR-VIS face recognition datasets, which means, different facial images are synthesized to fit different target datasets. Such dataset-specific face generation degrades the generalizability of the NIR-VIS face recognition networks.

To tackle the above problems, we propose a novel physically-based facial image generation method, where high-quality NIR-VIS facial image pairs are generated based on acquired renderable 3D facial assets. Rendering photorealistic 3D facial datasets enables us to acquire paired labelled training data with controllable identity, pose, expression and illumination. In contrast with generative methods, the rendered identity does not change at all when varying other parameters, which greatly aids training. However, acquiring human rendering assets requires tremendous manually work, either by scanning systems [37, 18, 27, 28] or artists [47]. The available datasets are either small in size [2] or do not contain relightable reflectance [51], such as the diffuse albedo, specular albedo and normals. Recent works [29, 30, 36, 10] have introduced methods that produce high-quality renderable assets from arbitrary facial images. Moreover, Wood *et al.* [47] showed that high-quality synthetic facial data, can be successfully used for computer vision tasks, including landmark localization and facial parsing. However, to our knowledge, there exists no dataset or method capable of producing renderable 3D faces, in both VIS and NIR domains.

Utilizing a state-of-the-art facial reflectance acquisition method [30], we generate numerous such facial assets and transform them from VIS to NIR, and then render both under the same conditions, in order to generate high-quality training data. Since our novel transformation method is applied per-pixel on high-resolution reflectance maps, the identity of the subject is perfectly preserved in both NIR and VIS. Faces generated by the proposed methods are illustrated in Fig. 1(b). As can be seen, our NIR-VIS face generation outperforms [13] (Fig. 1(a)) in preserving identity consistency as well as retaining the diversity of facial appearances.

The generated high-quality NIR-VIS facial image dataset is then used, along with a VIS face recognition dataset, to train the NIR-VIS face recognition network. To facilitate the identity feature learning as well as reduce the modality discrepancy, an IDentity-based Maximum Mean Discrepancy (ID-MMD) loss is proposed, which pulls the feature centroids of the same identity in the NIR domain and the VIS domain closer. Assisted by the ID-MMD loss, the gap between NIR images and VIS ones is bridged at the domain level, and meanwhile, the network is encouraged to focus on identity

features rather than facial details of instances, such as poses and accessories. Overall, our main contributions can be summarized as:

- A method capable of generating vast amounts of paired NIR and VIS facial images, of various identities, poses and illumination via 3D facial reconstruction and a novel VIS-to-NIR transformation for facial reflectance, is proposed.
- To bridge the gap between the NIR images and VIS images, we propose an IDentity-based Maximum Mean Discrepancy (ID-MMD) loss, which not only reduces the modality discrepancy at the domain level but encourages the network to attend to identity features instead of facial details.
- Extensive experiments on four NIR-VIS face recognition benchmarks demonstrate that the proposed method achieves comparable performance with the state-of-the-art methods, without requiring any existing NIR-VIS face recognition dataset. By slightly fine-tuning the models on the target NIR-VIS face recognition datasets, our method surpasses the SOTA performance.

## 2 Background and Related Work

**NIR-VIS Face Recognition.** To facilitate the NIR-VIS face recognition, earlier works focuses on learning modality-invariant features [21, 22, 24, 23]. [21] extracts features for NIR and VIS images via a shared feature extractor pretrained on large-scale VIS face images, where the modality-invariant identity information is then filtered out by an orthogonal constraint. Similarly, DFAL [24] and OMDRA [23] attempt to purify the identity information via decoupling identity-related representations from modality-invariant ones. To further reduce the gap between two modalities, WCNN [22] minimizes the Wasserstein distance between feature distributions of NIR and VIS images. WIT [41] treats each modality as a whole and squeezes the centers of the two modalities via a center maximum mean discrepancy loss. SMCL [46] designs a center-based loss to regulate the relationship between the syncretic modality and the NIR(VIS) one. However, due to the limited amount of NIR-VIS data, the aforementioned methods are generally in the mire of the over-fitting problem.

To address the problem, generative models are involved to facilitate the NIR-VIS face recognition task by transferring VIS face images to NIR ones via the image-to-image translation [31, 40, 54] or synthesizing heterogeneous face image pairs [13]. [31] produces VIS faces from NIR images via learning the patch-to-patch mapping between NIR-VIS image pairs. [40] transfers NIR face images to the corresponding VIS ones via a two-path Generative Adversarial Networks (GAN)-based framework. [54] proposes a multi-stream feature-level fusion technique based on GAN to synthesize visible images from polarimetric thermal images. However, the improvement brought by the "one-to-one" face synthesis strategy is limited by the number of images within the NIR-VIS face recognition datasets, as well as the lack of the attribute diversity of generated images. Henceforth, instead of adopting the conditional image generation, [13] employs the Variational AutoEncoders (VAE) to synthesize a paired NIR-VIS face images for a given new identity. The identity representations are obtained from an identity sampler, which is pre-trained on a large-scale face recognition dataset. Inspired by [13], generating a large number of paired NIR-VIS face images is beneficial to the NIR-VIS face recognition. However, we notice that, when generating multiple NIR-VIS face image pairs from a given identity representation via [13], the identity consistency is not well preserved. Such disadvantage hinders the potential of generated face images to boost the recognition performance since they cannot provide an explicit guidance for identity features learning. Therefore, we propose a novel NIR-VIS face image generation method, where a physically-based renderer is used to generate a vast photo-realistic dataset.

**NIR-VIS Rendering.** Synthetic datasets have long been used in face analysis problems with moderate success [55, 5, 15, 14, 39]. DA-GAN [55] and UV-GAN [5] pioneered realistic profile face generation for pose-invariant face recognition. A recent work by Wood *et al.* [47], achieves state-of-the-art performance in various facial analysis tasks, by using hand-crafted high-quality photo-realistic facial avatars. 3D facial appearance methods [29, 30, 36, 10] leverage deep generative models and differentiable rendering to reconstruct facial assets that can be photo-realistically rendered. Moreover, another work [3] projects incomplete renderings to the latent space of a deep generative model and reconstructs corresponding photo-realistic facial images. Although potent in the VIS spectrum, such methods are not directly usable in NIR.

The literature around NIR rendering from VIS assets remains limited. Wu *et al.* [50] introduced a wavelength-dependent Bidirectional Reflectance Distribution Function (BRDF) for Mid-Wavelength Infrared (MWIR) landscape scene simulation. Moreover, Aguerre *et al.* [1] simulate urban thermal imaging. While closest to our work, none of the above can directly render photo-realistic faces in NIR-VIS. In that manner, we employ the recent works in photo-realistic synthetic face and combine it with our novel VIS-to-NIR transformation to unlock the potential for NIR-VIS facial matching.

## 3    Proposed Method

In this work, in order to overcome the over-fitting problem caused by the available small-scale NIR-VIS face recognition datasets, we generate a vast, high-resolution and photo-realistic dataset consisting of large-scale identities with various poses and illumination in the NIR and VIS spectra (Sec. 3.1). Then, the generated NIR-VIS facial image pairs, along with a large-scale VIS face recognition dataset, are used to train the NIR-VIS face recognition network, without the aid of any existing NIR-VIS datasets (Sec. 3.2).

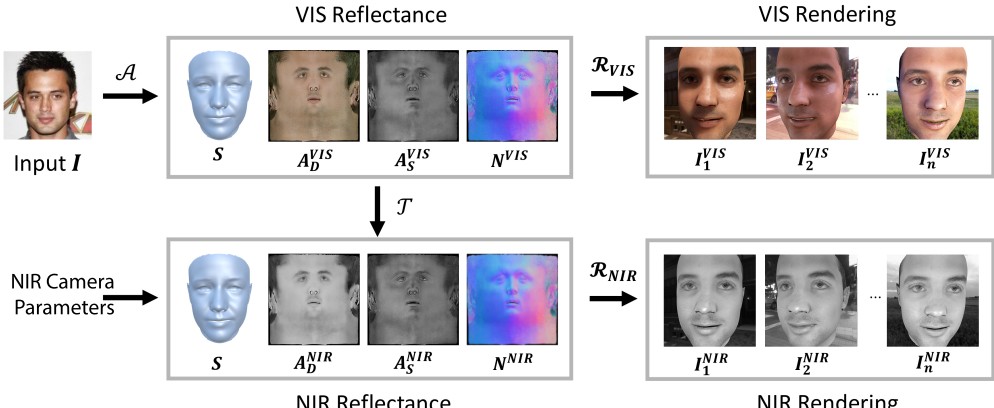

Figure 2: We generate paired NIR-VIS facial images by a) acquiring facial VIS reflectance from the public dataset of facial images ($\mathcal{A}$), b) transforming the VIS reflectance maps to NIR ($\mathcal{T}$) by exploiting the way human skin reacts to different spectra and c) rendering both VIS and NIR facial images in various poses and illumination conditions with the physically-based $\mathcal{R}_{VIS}$ and $\mathcal{R}_{NIR}$ renderers, that model a common VIS camera, and a NIR camera with a flood illuminator.

### 3.1    NIR-VIS Facial Rendering

We introduce a method able to generate vast amounts of photo-realistic pairs of NIR-VIS face images, to facilitate the heterogeneous face recognition task. Instead of using 2D generative models, we reconstruct a set of 3D facial assets capable of photo-realistic rendering in the VIS domain. The assets include a facial shape, and VIS spatially-varying reflectance attributes, encoded in UV-space texture maps, which can be sampled during rendering [30]. Meanwhile, we transform these into the NIR domain with a novel VIS-NIR transformation method and render paired images, as described in the following sections. In this manner, we can create paired labelled NIR-VIS facial images, of numerous identities, in arbitrary poses and illumination conditions.

**VIS Reflectance Data Acquisition.** Recent works [29, 30, 36, 10] have achieved high-quality render-ready VIS face reconstructions, from arbitrary faces. In that manner, we acquire CelebA [35], a dataset of over 200k facial images $\mathbf{I}_i$, and fit a textured 3D Morphable Model (3DMM), a GANFIT-based [16] fitting $\mathcal{F}$, to acquire a reconstructed texture and mesh $\mathbf{S}_i$. We employ an image-to-image translation network $\mathcal{A}$, based on AvatarMe++ [30], that disentangles facial reflectance maps from the reconstructed 3DMM texture maps. In that manner, for each image $\mathbf{I}_i$, we acquire the shape $\mathbf{S}_i$, diffuse albedo $\mathbf{A}_{\mathbf{D_i}}^{\mathbf{VIS}}$, specular albedo $\mathbf{A}_{\mathbf{S_i}}^{\mathbf{VIS}}$ and surface normals $\mathbf{N}_{\mathbf{i}}^{\mathbf{VIS}}$ (also referred to as specular normals [30]), in the VIS domain:

$$\left[\mathbf{S}_i, \mathbf{A}_{\mathbf{D_i}}^{\mathbf{VIS}}, \mathbf{A}_{\mathbf{S_i}}^{\mathbf{VIS}}, \mathbf{N}_{\mathbf{i}}^{\mathbf{VIS}}\right] = \mathcal{A}(\mathcal{F}(\mathbf{I}_i)) \tag{1}$$

**VIS to NIR Reflectance Transformation.** Human skin is a dielectric material and exhibits both diffuse and specular reflectance [37]. In the VIS spectrum, the diffuse albedo $\mathbf{A_D^{VIS}}$ describes the amount of light emitted per RGB channel when a medium is lit by uniform white illumination. On the other hand, the specular albedo $\mathbf{A_S^{VIS}}$ describes the intensity of the incoming illumination that is reflected, at the direction of the normals $\mathbf{N^{VIS}}$. To transform these spatially-varying reflectance values to the NIR spectrum, we define the following empirical model, based on the assumption that the reflectance attributes can be linearly described by the wavelength of the incident illumination, in the VIS $(380 - 700nm)$ and a NIR illumination $(850nm)$.

Surface normals can be acquired using a single-sensor input, R,G or B [37, 27]. It is well-established that for normals measured under white illumination, shorter wavelength of incident illumination exhibits sharper surface details, due to the subsurface scattering of light in skin [37, 27]. For a green wavelength $w^G$ and red-wavelength $w^R$, we calculate the width $\sigma$ of a Gaussian kernel $\mathcal{G}$ required to minimize the difference between the green normals $\mathbf{N^G}$ and red normals $\mathbf{N^R}$ (Eq. 2). Such transformations have been shown effective for normals manipulation [27, 28]. For a NIR wavelength $w^{NIR}$, we scale $\sigma$, based on the distance between the red and NIR wavelength and apply $\mathcal{G}$ on VIS normals $\mathbf{N^{VIS}}$ (in our case green channel [30], $\mathbf{N^{VIS}} = \mathbf{N^G}$) to acquire the NIR normals $\mathbf{N^{NIR}}$:

$$\mathbf{N^{NIR}} = \mathcal{G}\left(\mathbf{N^{VIS}}, \frac{w^{NIR} - w^G}{w^R - w^G}\sigma\right), \quad \sigma = \arg\min_{\sigma}\left|\mathbf{N^R} - \mathcal{G}(\mathbf{N^G}, \sigma)\right|_2 \qquad (2)$$

The NIR sensor is monochrome and its response is more similar to that of the VIS red channel. Given that a facial VIS diffuse albedo $\mathbf{A_D^{VIS}}$ is measured under uniform white light, its red channel $\mathbf{A_D^R}$ measures the skin's response to the red-wavelengths $w^R$. Assuming again a relationship between wavelength and spectral response, from red to infrared, we define the NIR diffuse albedo $\mathbf{A_D^{NIR}}$, as the blurred red channel albedo $\mathbf{A_D^R}$. In contrast to Eq. 2, here we use a Bilateral Filter [42], in order to preserve the edges of the facial details.

Finally, we retain the VIS specular albedo as $\mathbf{A_S^{VIS}} = \mathbf{A_S^{NIR}}$, assuming it is wavelength independent in the visible spectrum [37, 27]. However, we decrease the overall specular roughness, proportionally to the distance of the target NIR wavelength from the mean of the visible spectrum, following [50]. Following the above, we define the complete NIR-VIS transformation function as $\mathbf{A_D^{NIR}}, \mathbf{A_S^{NIR}}, \mathbf{N^{NIR}} = \mathcal{T}(\mathbf{A_D^{VIS}}, \mathbf{A_S^{VIS}}, \mathbf{N^{VIS}}, w^{NIR})$.

**Paired NIR-VIS Rendering.** The importance of explicitly extracting detailed albedo and normal maps, lies in the fact that we can employ photo-realistic rendering algorithms, such as GGX [44] in our case. We collect a set of $n$ environment maps $\mathbf{E_1}, \ldots, \mathbf{E_n} \in \mathcal{E}$, that define incoming illumination of various realistic scenes [4]. We define a VIS physically-based renderer $\mathcal{R}^{VIS}(\mathbf{S}, \mathbf{R^{VIS}}, \mathbf{M}, \mathbf{E}) \rightarrow \mathbf{I} \in \mathbb{R}^{h \times w \times 3}$, where $h, w$ is the size of the rendered image, $\mathbf{S}$ is the facial shape, $\mathbf{R^{VIS}} = \left[\mathbf{A_D^{VIS}}, \mathbf{A_S^{VIS}}, \mathbf{N^{VIS}}\right]$ is the reflectance, $\mathbf{M}$ is a rotation matrix for the shape and $\mathbf{E}$ is an environment map. For NIR rendering, we define a similar but monochrome renderer $\mathcal{R}^{NIR}(\mathbf{S}, \mathbf{R^{NIR}}, \mathbf{M}, \mathbf{E}) \rightarrow \mathbf{I} \in \mathbb{R}^{h \times w \times 1}$. Finally, as NIR sensors typically rely on a flood illuminator placed adjacent to the lens, we create an environment map $\mathbf{E_f}$ with only a flood illuminator placed in the camera direction. Then, for a map $\mathbf{E_i}$, we define the equivalent NIR map as $\mathbf{E_i^{NIR}} = \mathbf{E_i} + \mathbf{E_f}$.

We then use the generated VIS and NIR facial assets and the VIS and NIR renderers, to generate sets of paired NIR-VIS facial images, under arbitrary illumination and pose. For a given subject $i$ with shape $\mathbf{S_i}$ and reflectance $\mathbf{R_i^{VIS}}$ and $\mathbf{R_i^{NIR}}$, we randomly sample a) an environment map $\mathbf{E_j} \in \mathcal{E}$, which is rotated along the Y axis by a random angle $\theta_j \in [0, 360]$, and b) a rotation matrix $\mathbf{M_j}$. A NIR-VIS image pair is rendered by $\mathbf{I}_{i,j}^{VIS} = \mathcal{R}_i^{VIS}(\mathbf{S_i}, \mathbf{R_i^{VIS}}, \mathbf{M_j}, \mathbf{E_j})$ and $\mathbf{I}_{i,j}^{NIR} = \mathcal{R}_i^{NIR}(\mathbf{S_i}, \mathbf{R_i^{NIR}}, \mathbf{M_j}, \mathbf{E_j^{NIR}})$.

## 3.2 NIR-VIS Face Recognition

As a departure from most NIR-VIS face recognition works, the generated heterogeneous facial dataset is employed, along with a large-scale VIS face recognition dataset, to train a NIR-VIS face recognition network explicitly. Specifically, we present the large-scale VIS face recognition dataset containing $C$ identities as $X = \{x_i\}_{i=1}^{N}$, and the corresponding identity label as $Y = \{y_i\}_{i=1}^{C}$. Similarly, the synthesized NIR-VIS dataset with $C_s$ identities is denoted as $X_s = \{x_i\}_{i=1}^{N_s}$ with label $Y_s = \{y_i\}_{i=1}^{C_s}$. $N$ and $N_s$ denote the number of images in the VIS dataset and the synthesized

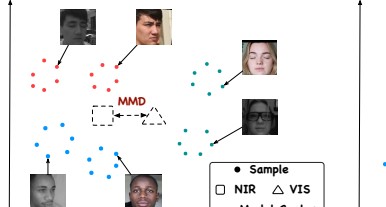 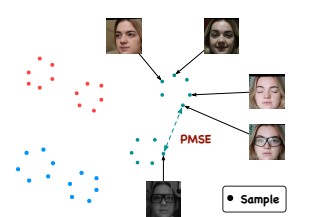 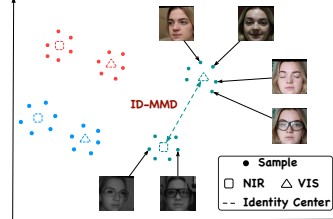

Figure 3: The comparisons between the traditional MMD loss, the Pairwise Mean Square Error (PMSE) and the proposed IDentity-based MMD (ID-MMD) loss. Different identities are represented by different colors (best viewed in color).

NIR-VIS dataset, respectively. Note that, since the synthesized identities are from CelebA [35], there are no overlapped identities between the VIS face recognition dataset and the synthesized one. Given the final training set built by $X$ and $X_s$, following [13], a widely-used face recognition network [48] is trained under the supervision of the identity loss [6] and the proposed IDentity-based Maximum Mean Discrepancy (ID-MMD) loss. The NIR-VIS face recognition task is conducted with the identity features derived from the network.

**Identity Loss.** To improve the discrimination power of the face recognition network, we employ the margin-based softmax loss [6, 45, 33] $\mathcal{L}_{id}$ during training, which is denoted as follows,

$$\mathcal{L}_{id} = -\frac{1}{N_r + N_s} \sum_i \log \frac{e^{s(\cos(m_1\theta_{y_i,i}+m_2)-m_3)}}{e^{s(\cos(m_1\theta_{y_i,i}+m_2)-m_3)} + \sum_{j \neq y_i} e^{s\cos\theta_{j,i}}}, \tag{3}$$

where $cos\theta_{j,i} = W_j^T f_i$, $f_i$ is the normalized feature of $i$-th image, and $W_j$ is the normalized weight vector of the $j$-th class. $\theta_{j,i}$ refers to the angle between $W_j$ and $f_i$. $m_1$, $m_2$, and $m_3$ is the margin parameters. $s$ is the feature scale. $N_r$ and $N_s$ are the training sample numbers of real VIS faces and synthesized NIR-VIS faces.

**ID-MMD Loss.** To overcome the main challenge of the NIR-VIS face recognition task, *i.e.,* the cross-modality discrepancy, the Maximum Mean Discrepancy (MMD) loss [17] designed for the transfer learning task is adopted. Formally, given a mini-batch containing $M$ NIR images and $N$ VIS images, the MMD loss $\mathcal{L}_{mmd}$ is formulated as follows,

$$\mathcal{L}_{mmd} = \left\| \frac{1}{M} \sum_{i=1}^{M} \phi(x_i^{nir}) - \frac{1}{N} \sum_{j=1}^{N} \phi(x_j^{vis}) \right\|_{\mathcal{H}}^2, \tag{4}$$

where $\phi(\cdot)$ represents the kernel function that maps the original data to a Reproducing Kernel Hilbert Space (RKHS) $\mathcal{H}$. Although the MMD loss reduces the domain discrepancy by aligning NIR-VIS feature distributions, rigidly adopting such domain-level constraint to the NIR-VIS face recognition network training is sub-optimal, since it considers each modality as a whole and ignores the identity feature distribution within the modality, as shown in Fig 3.

To solve the problem, an explicit solution is reducing the distance between each NIR-VIS image pair of the same identity in the latent space, *i.e.,* minimizing the Pairwise Mean Square Error (PMSE) $\mathcal{L}_{pmse}$ between the NIR-VIS features. Concretely, random sampling $P$ identities from the heterogeneous dataset, and for each identity, sampling $K$ NIR images and $K$ VIS images, to form the mini-batch with $2 \times PK$ images. The $\mathcal{L}_{pmse}$ is denoted as follows,

$$\mathcal{L}_{pmse} = \frac{1}{P}\frac{1}{K} \sum_{p=1}^{P} \sum_{k=1}^{K} \left\| f_{p,k}^{nir} - f_{p,k}^{vis} \right\|^2, \tag{5}$$

where $f_{p,k}^{nir/vis}$ denotes the normalized feature of $k$-th NIR/VIS image of $p$-th identity. Although the identity distribution is considered, such pairwise loss reduces the modality discrepancy at the instance level, where the network is highly likely to focus on facial details, such as poses and accessories, rather than identity features. Taking the girl wearing glasses in Fig. 3 as an example, the PMSE loss may reduce the feature distance between the NIR-VIS image pair by encouraging the network to attend to the frontal pose or the glasses, instead of the identity features.

To address the problems of the domain-based (MMD loss) and the instance-based (PMSE) modality discrepancy reduction loss, we propose an ID-based MMD loss $\mathcal{L}_{idmmd}$, which bridges the modality

gap by reducing the distance between the NIR-VIS feature centroids of the same identity in the RKHS. Formally, for the given mini-batch, the proposed $\mathcal{L}_{idmmd}$ is denoted as follows,

$$\mathcal{L}_{idmmd} = \frac{1}{P} \sum_{p=1}^{P} \left\| \phi(\frac{1}{K} \sum_{k=1}^{K} f_{p,k}^{nir}) - \phi(\frac{1}{K} \sum_{k=1}^{K} f_{p,k}^{vis}) \right\|_{\mathcal{H}}^2 . \tag{6}$$

The proposed identity-based modality discrepancy reduction loss not only reduces the modality gap between NIR and VIS images, but also encourages the features of the same identity within each modality to be compactly distributed, *i.e.,* reducing the intra-modality discrepancy. Overall, the objective of the NIR-VIS face recognition network is denoted as $\mathcal{L} = \mathcal{L}_{id} + \lambda * \mathcal{L}_{idmmd}$, where $\lambda$ is the balancing parameter set as 100 during training.

## 4 Experiments

### 4.1 Databases and Protocols

Four NIR-VIS face recognition datasets are used to evaluate the proposed method. Specifically, the CASIA NIR-VIS 2.0 [32] (725 identities) and the LAMP-HQ [52] (573 identities) are the most challenging NIR-VIS face recognition datasets due to the huge facial appearance diversity in poses, illumination, and ages. Following [32, 52], the ten-fold experiments are conducted on both datasets. For each fold, approximately 50% identities are randomly selected as the training set and the rest are adopted as the testing set. Note that, there is no overlap between the training and testing sets. Following [13, 52], the verification rate (VR)@false accept rate (FAR)=0.01%, VR@FAR=0.1%, and the Rank-1 accuracy, are employed for evaluation. In terms of two low-shot NIR-VIS face recognition datasets: the Oulu-CASIA NIR-VIS [26] and the BUAA-VisNir [25], the identities within the datasets are split as 20/20 and 50/100, respectively, for the setting of the training/testing set. Considering the small data scale, the VR@FAR=0.1% and the Rank-1 accuracy are adopted as the evaluation metrics.

### 4.2 Experimental Details

**NIR-VIS Face Generation.** To acquire the 3D facial assets, we use AvatarMe++[30], trained with RealFaceDB [29] at textures of $1024 \times 1024$ pixels, on a GANFIT [16] base. The dataset used is CelebA [35], however, other 2D facial datasets can be used to extend generalization. For the rendering, we employ highly parameterizable commercial rendering software Marmoset Toolbag [38]. For each identity from CelebA, we synthesize 20 VIS and NIR facial image pairs, under different poses and illumination.

**NIR-VIS Face Recognition.** Following [13], we utilize LightCNN-29 [48] as the NIR-VIS face recognition backbone. To make a fair comparison with [13], which adopts about 5 million images from the MS-Celeb-1M [20] dataset for pre-training, we use a subset from the large-scale VIS dataset [56], *i.e.,* WebFace4M [57], for training. WebFace4M contains 4 million images of 200k randomly chosen identities from WebFace260M [57]. Additionally, instead of using $128 \times 128$ facial images as inputs, all the face images are aligned and cropped to $112 \times 112$ [7, 19] in the paper.

During training, we first train the network with identity loss $\mathcal{L}_{id}$ for 20 epochs on the WebFace4M and the synthesized dataset. Then, we fine-tune the network on the synthesized images with both identity loss $\mathcal{L}_{id}$ and ID-MMD loss $\mathcal{L}_{idmmd}$ for 5 epochs. The batch size is set as 512. During fine-tuning, 32 identities are randomly selected to form a mini-batch, and for each identity, 8 VIS and 8 NIR images are randomly selected. Stochastic gradient descent (SGD) is adopted as the optimizer, where the momentum is set to 0.9 and the weight decay is set to $1e$-4. The learning rate is set to $1e$-2 initially and decays by a factor of 0.5 per 10 epochs.

### 4.3 Ablation Study

**NIR Reflectance Generation.** We show the importance of each component in our proposed NIR reflectance generation method, by reconstructing subjects from LAMP-HQ [52] and showing the effectiveness of our algorithm against alternative approaches. In Fig. 4 and Tab. 1, we compare our method with monochrome rendering of VIS assets or remove some of our transformations. Not only

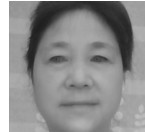 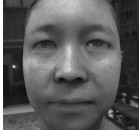 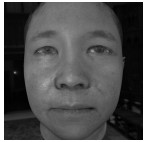 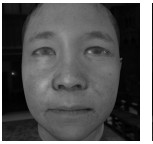 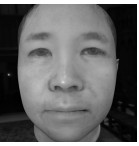 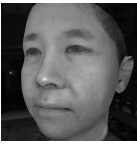 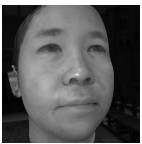

(a) GT [52]  (b) Gr. $\mathcal{R}^{\mathcal{VIS}}$  (c) $\mathcal{R}^{\mathcal{NIR}}$  (d) +$\mathbf{N^{NIR}}$  (e) Ours, front (f) Ours, side (g) Ours, side

Figure 4: From left to right: a) Ground Truth sample from LAMP-HQ[52], b) grayscale rendering with VIS Renderer $\mathcal{R}^{\mathcal{VIS}}$ and VIS assets, c) rendering with NIR Renderer $\mathcal{R}^{\mathcal{NIR}}$ of VIS assets with $\mathbf{E^{NIR}}$, d) NIR Renderer $\mathcal{R}^{\mathcal{NIR}}$ with our NIR-transformed normals $\mathbf{N^{NIR}}$, e) our complete method, including $\mathcal{R}^{\mathcal{NIR}}$, $\mathbf{N^{NIR}}$ and our transformed diffuse albedo $\mathbf{A_D}^{NIR}$ and f), g) additional poses.

our method is capable of producing photo-realistic NIR renderings, but can do is in various poses while preserving the subject identity, since it is based on rendering 3D assets.

|  | Gray $\mathcal{R}^{\mathcal{VIS}}$ | $\mathcal{R}^{\mathcal{NIR}}$ | +$\mathbf{N^{NIR}}$ | Ours |
|---|---|---|---|---|
| PSNR | 16.61 | 17.32 | 17.34 | **21.03** |
| MSE | 0.021 | 0.018 | 0.018 | **0.007** |

| Method | MS↑ | | MIS↓ | | FID↓ |
|---|---|---|---|---|---|
|  | 1v1 | 1vN | VIS-VIS | NIR-VIS |  |
| DVG-Face | 0.429 | 0.244 | 0.149 | 0.138 | 0.877 |
| Ours | **0.641** | **0.411** | **0.099** | **0.084** | **0.609** |

Table 1: NIR rendering ablation study, showing average MSE and PSNR, for LAMP-HQ [52] reconstructions with our method and the ablation alternatives shown in Fig. 4.

Table 2: Comparisons of 1) the identity consistency (MS), 2) the identity diversity (MIS), and 3) the distribution distance between generated and real data (FID) on LAMP-HQ.

**Identity Consistency and Diversity.** Following DVG-Face [13], we analyze the identity consistency, the identity diversity and the distribution consistency of the generated NIR-VIS images via evaluation metrics - Mean Similarity (MS), Mean Instance Similarity (MIS), and Frechet Inception Distance (FID), respectively. To make a fair comparison, we randomly select 1000 identities from NIR-VIS images generated by DVG-Face and ours. For each identity, 16 NIR images and 16 VIS images are randomly selected for evaluation. The results are reported in Tab. 2. Note that, apart from measuring MS between each NIR-VIS image pair as DVG-Face, we also measure the feature similarity across multiple images generated for a given identity, which are indicated by $1v1$ and $1vN$ in Tab. 2, respectively. The higher intra-class (identity) similarity (MS) proves the superiority of our generation in preserving identity consistency. Additionally, our method achieves a lower inter-class similarity (MIS), which ensures identity diversity. Meanwhile, a lower FID evaluated by LightCNN [48, 13] facilitates the adaptation to real-world NIR-VIS face recognition datasets.

**Effectiveness of Generated Data.** To prove the proposed NIR-VIS face generation method can significantly facilitate the NIR-VIS face recognition, we compare the performances of models trained with the proposed ID-MMD loss under different percentages $\{0\%, 10\%, 50\%, 100\%\}$ of generated data. In Tab. 3, the test results on LAMP-HQ show that the generated images continuously contribute to performance improvement and the best result is achieved when all generated data are involved.

**Comparisons of Domain Adaption Losses.** To compare our ID-MMD loss with other modality discrepancy reduction losses, the ablation studies are conducted on the LAMP-HQ dataset [52]. Specifically, PMSE loss, MMD loss, and ID-MMD loss are employed to supervise the learning of the NIR-VIS face recognition network. The corresponding recognition performances are reported in Tab. 4. It can be observed that the network achieves the best performance when optimizing by the proposed ID-MMD loss, surpassing the PMSE loss by 1.56% and the MMD loss by 1.04% when VR@FAR=0.01%.

| Ratio | FAR=0.01% | FAR=0.1% | Rank-1 |
|---|---|---|---|
| 0% | 67.28 ± 1.9 | 85.26 ± 1.0 | 96.12 ± 0.2 |
| 10% | 82.13 ± 1.7 | 92.87 ± 0.8 | 97.85 ± 0.2 |
| 50% | 90.64 ± 1.5 | 97.18 ± 0.4 | 98.64 ± 0.2 |
| 100% | **91.97 ± 1.5** | **97.96 ± 0.3** | **98.87 ± 0.3** |

|  | FAR=0.01% | FAR=0.1% | Rank-1 |
|---|---|---|---|
| $\mathcal{L}_{\mathrm{idmmd}}$ | **91.97 ± 1.5** | **97.96 ± 0.3** | **98.87 ± 0.3** |
| $\mathcal{L}_{\mathrm{mmd}}$ | 90.93 ± 1.6 | 97.27 ± 0.3 | 98.04 ± 0.3 |
| $\mathcal{L}_{\mathrm{pmse}}$ | 90.41 ± 1.5 | 96.88 ± 0.2 | 97.85 ± 0.3 |

Table 3: Performance comparisons between NIR-VIS face recognition models trained with 0%, 10%, 50%, 100% generated data on the LAMP-HQ dataset.

Table 4: The comparisons between the performances of the backbone network (LightCNN-29) trained with different modality discrepancy reduction losses on the LAMP-HQ dataset.

**Effectiveness of IDMMD.** To better understand the advantage of the generated NIR-VIS facial images as well as the proposed IDentity-based Maximum Mean Discrepancy (ID-MMD) loss, we visualize the identity feature distributions of the Oulu-CASIA NIR-VIS [26] and the LAMP-HQ [52]

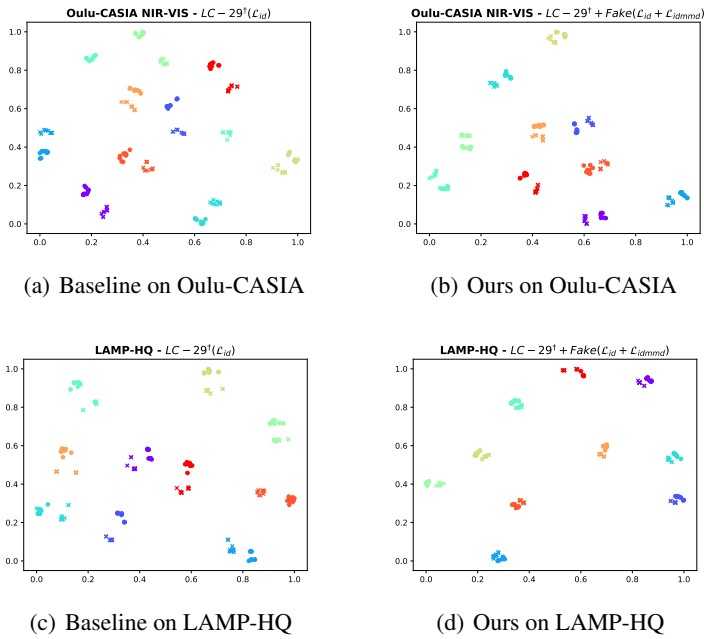

(a) Baseline on Oulu-CASIA

(b) Ours on Oulu-CASIA

(c) Baseline on LAMP-HQ

(d) Ours on LAMP-HQ

Figure 5: t-SNE [43] visualization of features of 10 identities randomly selected from Oulu-CASIA NIR-VIS, and LAMP-HQ. Different identities are denoted by different colors. •: NIR images; ×: VIS images. (Best viewed in color).

dataset. Specifically, for each dataset, we randomly select 10 identities from the testing set. For each identity, we randomly selected 10 NIR images and 10 VIS images. We visualize the distribution of features derived by the baseline network "$LC - 29^{\dagger}(\mathcal{L}_{id})$" and the proposed one "$LC - 29^{\dagger}$ + Fake $(\mathcal{L}_{id} + \mathcal{L}_{idmmd})$" with t-SNE [43]. The visualization results are demonstrated in Fig. 5. Different identities are denoted by different colors. As can be seen, after involving the generated NIR-VIS facial images and the ID-MMD loss, the NIR-VIS features of the same identity are pulled closer. Meanwhile, for each identity, the features within the NIR/VIS domain distribute more compactly. Such visualization results suggest that the proposed method can effectively reduce both intra-modality and inter-modality discrepancies.

### 4.4 Comparisons with State-of-the-art Methods

We extensively compare our method with the state-of-the-art (SOTA) methods on four NIR-VIS face recognition benchmarks. The performances are reported in Tab. 5. Following [13], we set the NIR-VIS face recognition baseline LightCNN-29 [48]. Unlike [13], our baseline model, 1) is trained on the WebFace4M [57], 2) takes $112 \times 112$ face images as input, and 3) is trained under the margin-based softmax loss [6]. As can be seen in Tab. 5, our baseline models achieve a comparable performance with [48] at a lower input resolution.

We show that, with the aid of the synthesized NIR-VIS face images and the modality discrepancy reduction loss, our method greatly improves the baseline performances on four benchmark datasets. To intuitively illustrate the effectiveness of the proposed method, we visualize the feature similarity distribution of positive pairs (belonging to the same identity) and negative pairs (belonging to different identities) of LAMP-HQ in Fig. 6. Benefiting from the generated data and ID-MMD loss, the similarities between positive pairs increase while the similarities between negative pairs decrease.

As can be seen, without requiring any existing NIR-VIS face recognition datasets, our method has achieved comparable performances with SOTA methods on the CASIA NIR-VIS 2.0 and the Oulu-CASIA NIR-VIS datasets, and has surpassed the SOTA performance by a large margin on the other two. Especially, on the challenging LAMP-HQ dataset, our method outperforms the SOTA one [52] by 19.8% in terms of VR@FAR=0.1%.

The performances can be further boosted after slightly fine-tuning the models on the target datasets. Specifically, as reported in the last two rows of Tab. 5, by adopting the identity loss ($\mathcal{L}_{id}$) during

| Method | CASIA NIR-VIS 2.0 | | | LAMP-HQ | | | Oulu-CASIA NIR-VIS | | BUAA-VisNir | |
|---|---|---|---|---|---|---|---|---|---|---|
| | FAR=0.01% | FAR=0.1% | Rank-1 | FAR=0.01% | FAR=0.1% | Rank-1 | FAR=0.1% | Rank-1 | FAR=0.1% | Rank-1 |
| TRIVET [34] | $74.5 \pm 0.7$ | $91.0 \pm 1.3$ | $95.7 \pm 0.5$ | - | - | - | 33.6 | 92.2 | 80.9 | 93.9 |
| IDR [21] | - | $95.7 \pm 0.7$ | $97.3 \pm 0.4$ | - | - | - | 46.2 | 94.3 | 84.7 | 94.3 |
| W-CNN [22] | $94.3 \pm 0.4$ | $98.4 \pm 0.4$ | $98.7 \pm 0.3$ | - | - | - | 54.6 | 98.0 | 91.9 | 97.4 |
| ADFL [40] | - | $97.2 \pm 0.5$ | $98.2 \pm 0.3$ | - | $73.3 \pm 2.2$ | $95.1 \pm 0.5$ | 60.7 | 95.5 | 88.0 | 95.2 |
| RCN [9] | - | $98.7 \pm 0.2$ | $99.3 \pm 0.2$ | - | - | - | - | - | - | - |
| PCFH [53] | - | $97.7 \pm 0.3$ | $98.8 \pm 0.3$ | - | $75.1 \pm 1.8$ | $95.3 \pm 0.5$ | 86.6 | 100.0 | 92.4 | 98.4 |
| MC-CNN [8] | - | $99.3 \pm 0.1$ | $99.4 \pm 0.1$ | - | - | - | - | - | - | - |
| PACH [11] | - | $98.3 \pm 0.2$ | $98.9 \pm 0.2$ | - | $75.3 \pm 1.7$ | $95.4 \pm 0.5$ | 88.2 | 100.0 | 93.5 | 98.6 |
| DVR [49] | $98.6 \pm 0.3$ | $99.6 \pm 0.3$ | $99.7 \pm 0.1$ | - | - | - | 84.9 | 100.0 | 96.9 | 99.2 |
| DVG [12] | $98.8 \pm 0.2$ | $99.8 \pm 0.1$ | $99.8 \pm 0.1$ | - | - | - | 92.9 | 100.0 | 97.3 | 99.3 |
| DFAL [24] | - | $98.7 \pm 0.2$ | $99.1 \pm 0.2$ | - | - | - | 93.8 | 100.0 | 99.2 | 100.0 |
| OMDRA [23] | - | $99.4 \pm 0.2$ | $99.6 \pm 0.1$ | - | - | - | 92.2 | 100.0 | 99.7 | 100.0 |
| LAMP-HQ [52] | - | $98.2 \pm 0.2$ | $99.2 \pm 0.0$ | - | $78.2 \pm 3.0$ | $97.3 \pm 0.2$ | 89.0 | 100.0 | 93.4 | 98.8 |
| DVG-Face [13] | $99.2 \pm 0.1$ | $99.9 \pm 0.0$ | $99.9 \pm 0.1$ | - | - | - | 97.3 | 100.0 | 99.1 | 99.9 |
| LC-29 [48] | - | $97.4 \pm 0.5$ | $98.1 \pm 0.4$ | - | $75.6 \pm 1.9$ | $94.6 \pm 0.3$ | 68.3 | 99.0 | 89.4 | 96.8 |
| LC$-29^{\dagger}$ ($\mathcal{L}_{id}$) | $85.0 \pm 1.9$ | $95.0 \pm 0.7$ | $96.2 \pm 0.7$ | $67.3 \pm 1.9$ | $85.3 \pm 1.0$ | $96.1 \pm 0.2$ | 85.1 | 100.0 | 96.7 | 99.0 |
| LC$-29^{\dagger}$ + Fake ($\mathcal{L}_{id}$) | $92.5 \pm 0.8$ | $98.6 \pm 0.4$ | $98.8 \pm 0.3$ | $84.9 \pm 1.6$ | $96.1 \pm 0.4$ | $98.4 \pm 0.3$ | 91.8 | 100.0 | 99.4 | 99.9 |
| LC$-29^{\dagger}$ + Fake ($\mathcal{L}_{id} + \mathcal{L}_{idmmd}$) | $97.5 \pm 0.5$ | $99.6 \pm 0.4$ | $99.6 \pm 0.2$ | $92.0 \pm 1.5$ | $98.0 \pm 0.3$ | $98.6 \pm 0.3$ | 94.5 | 100.0 | 99.8 | 100.0 |
| * LC-29 + Real($\mathcal{L}_{id}$) | $98.8 \pm 0.2$ | $99.7 \pm 0.2$ | $99.8 \pm 0.1$ | $94.5 \pm 0.8$ | $98.7 \pm 0.4$ | $99.0 \pm 0.3$ | 95.2 | 100.0 | 99.8 | 100.0 |
| * LC-29 + Real($\mathcal{L}_{id} + \mathcal{L}_{idmmd}$) | $\mathbf{99.9 \pm 0.1}$ | $\mathbf{100.0 \pm 0.0}$ | $\mathbf{99.9 \pm 0.1}$ | $\mathbf{98.6 \pm 0.4}$ | $\mathbf{99.4 \pm 0.3}$ | $\mathbf{99.1 \pm 0.3}$ | **99.1** | **100.0** | **99.8** | **100.0** |

Table 5: Comparisons with the state-of-the-art NIR-VIS face recognition methods on the CASIA NIR-VIS 2.0, LAMP-HQ, Oulu-CASIA NIR-VIS, and BUAA-VisNir datasets. LC-29: Adopting the network structure of LightCNN-29. †: Our baseline model. Fake: The synthesized images are included during training. $\mathcal{L}_{id}$, $\mathcal{L}_{idmmd}$: The objective(s) used during training/fine-tuning. *: Fine-tuning models on the target NIR-VIS datasets.

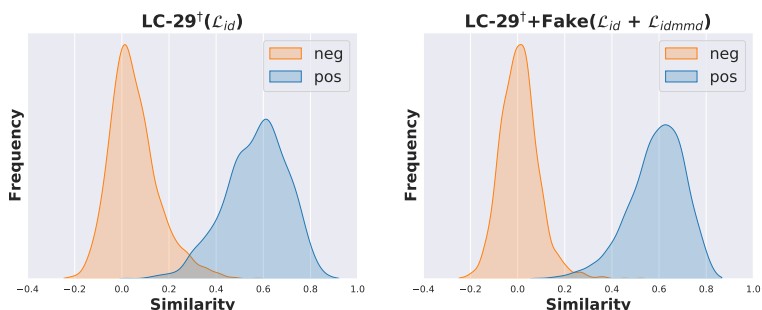

Figure 6: Feature similarity distribution of positive/negative pairs of LAMP-HQ.

fine-tuning, VR@FAR=0.01% increases from 92.0% to 94.5% on LAMP-HQ. After imposing the proposed ID-MMD loss ($\mathcal{L}_{idmmd}$) during the fine-tuning on target NIR-VIS face recognition datasets, the performances are further boosted. Concretely, an improvement of 6.6% on the VR@FAR=0.01% can be observed on LAMP-HQ. Additionally, on the two low-shot NIR-VIS face recognition datasets that fewer identities are contained, *i.e.,* the Oulu-CASIA NIR-VIS and BUAA-VisNir datasets, we surpass the DVG-Face [13] by 1.8% and 0.8% in terms of VR@FAR=0.1%, respectively. In summary, after fine-tuning on the target NIR-VIS face recognition datasets, our method outperforms all other competitors on four benchmarks.

## 5 Conclusion

To address the problem of insufficient NIR-VIS data for the cross-modality face recognition network training, this paper proposes a novel NIR-VIS face generation method, which enables the generation of vast amounts of photo-realistic paired NIR-VIS facial images with various poses and illumination while preserving the identity consistency. Such merit enables the use of the generated dataset, along with a large-scale VIS face recognition dataset, to train the NIR-VIS face recognition network, which can achieve comparable performance with the state-of-the-art methods without requiring any existing NIR-VIS face recognition datasets. Additionally, to bridge the domain gap between NIR images and VIS images during training, an IDentity-based Maximum Mean Discrepancy (ID-MMD) loss is proposed, which reduces the modality discrepancy at the domain level and encourages the network to focus on identity features rather than facial details. Qualitative and quantitative experiment results on four NIR-VIS face recognition benchmarks show the superiority of the proposed method. Finally, understanding the **social impacts** of facial matching, our method is modeled for and limited to mobile phone NIR sensors, and thus its use is aimed at a more user-friendly experience for such devices.

**Acknowledgements.** Stefanos Zafeiriou acknowledges support from the EPSRC Fellowship DE-FORM (EP/S010203/1) and a Google Faculty Fellowship.

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
