# OpenReview forum: "Physically-Based Face Rendering for NIR-VIS Face Recognition"
_NeurIPS.cc/2022/Conference — NeurIPS 2022 Accept_

### Official Review · Reviewer_wS34 · 2022-07-11

**Rating:** 5
**Confidence:** 4
**Soundness:** 3 good
**Presentation:** 3 good
**Contribution:** 3 good

**Summary:**

In this paper, the authors propose a method to synthesize near-infrared faces by transforming them from visible faces. Based on this, the authors can conduct NIR-VIS face recognition without any existing NIR-VIS face datasets. Besides, they also propose an Identity-based Maximum Mean Discrepancy loss to facilitate identity feature learning. The good performance shows the efficiency of their method.

**Questions:**

Please see my comments above.

**Limitations:**

The limitations and potential negative societal impact have been adequately addressed.

**Strengths And Weaknesses:**

Strengths:
1. The authors utilize a novel 3d-rendering based method to generate large-scale NIR-VIS paired data.
2. The authors propose an Identity-based Maximum Mean Discrepancy loss to facilitate identity feature learning.
3. The descriptions of implementations of the proposed synthesis method are detailed.

Weakness:
1. The motivation for using a 3d-rendering based generating dataset is not clearly illustrated, and it seems that the authors just utilize id-related loss when training. It is suggested to strengthen more use of the generation.
2. The comparison results are not sufficient. It is suggested to add generated data by different percentages to illustrate the effectiveness of the synthesis method.
3. I suggest more comparative visualizations to verify the effectiveness of the proposed method. In addition, the authors should discuss how generalizable the proposed method is in practical situations.

---

> ### Author Response · Authors · 2022-08-02
> **Response to Reviewer wS34**
>
> Thanks for the helpful suggestions, and we are open to further discussions.
> * Q1: The motivation for using a 3d-rendering based generating dataset is not clear, and it seems that the authors just utilise id-related loss when training. It is suggested to strengthen more use of the generation.
>
> As stated in Section 2 in the paper, the NIR-VIS face recognition task is in the mire of the over-fitting problem due to the limited amount of image pairs in current NIR-VIS face recognition datasets. To solve the problem, previous methods generally employ the generative models, such as Generative Adversarial Networks (GAN) [1] or Variational AutoEncoders (VAE) [2], to synthesis facial images. Specifically, GAN generates NIR face images from the corresponding VIS ones. However, the “one-to-one" face synthesis strategy (GAN) still suffers from the limited number of images in the NIR-VIS face recognition datasets. Although VAE solves the problem by synthesizing NIR-VIS face image pairs from identity representations, we notice the failure of preserving identity consistency when generating multiple NIR-VIS face image pairs from a given identity representation. To this end, we explore the generation of multiple paired facial images by photorealistically rendering in VIS and NIR 3D facial reconstructions, where the identity is perfectly preserved while changing the illumination and pose.
>
> Moreover, recent advances in facial reflectance reconstruction [3] enable us for the first time to acquire big datasets of facial shape and reflectance properties, which can be rendered in a fully controllable manner, in terms of illumination, background, pose and expression. The resulting vast datasets of labelled NIR-VIS pairs can be then used to augment the existing smaller-scale datasets used for training the model, clearly increasing the models' capabilities.
>
> As stated in the paper (Line 255-257), the synthesized images are employed throughout the NIR-VIS face recognition network training. Additionally, the proposed ID-MMD loss is not only id-related, but aims to reduce the domain discrepancy between the generated NIR and VIS images at each identity level.
> * Q2: It is suggested to add generated data by different percentages.
>
> Following the suggestion, experiments of adding generated data by different percentages are conducted on the LAMP-HQ dataset. Specifically, based on the baseline model LC-29${^\dagger}(L_{id})$, the generated data are added by the percentages of 10%, 50%, and 100%, respectively, during training. In the experiment, both identity loss and ID-MMD loss are used for network training. The comparison results are added as Table 3 in the revision.
>
> The model performances suggest that the generated images could continuously contribute to the performance improvements. Best performance is achieved when all generated data are involved.
>
> * Q3: More comparative visualisations. In addition, the authors should discuss how generalizable the proposed method is in practical situations.
>
> To better verify the effectiveness of the proposed method, we compare the mean cosine similarity between identity features of positive pairs and negative pairs of the LAMP-HQ dataset. Specifically, we randomly select 3k positive pairs (belonging to the same identity) and 3k negative pairs (belonging to different identities) from the test set. Then, identity features are extracted by model LC-29$^\dagger$ ($L_{id}$) and model LC-29$^\dagger$+Fake ($L_{id}$+$L_{idmmd}$) in the main paper, respectively. Here, we report the mean similarity of positive pairs and negative pairs as follows.
>
> Model | Positive | Negative
> :-: | :-: | :-:
> LC-29$^\dagger$ ($L_{id}$) | 0.569 | 0.053
> LC-29$^\dagger$+Fake ($L_{id}$+$L_{idmmd}$)| **0.590** | **0.009**
>
> As can be seen, as the generated data (Fake) and the ID-MMD loss ($L_{idmmd}$) are employed during training, the feature similarities between positive pairs increase while the similarities between negative pairs decrease. An intuitive visualization of the similarity distribution has been added as Fig. 5 in the revision.
>
> Our extensive experiments provide concrete evidence that, our method can achieve comparable performances with the state-of-the-art methods without requiring any existing NIR-VIS face recognition datasets, proving the generalizability of the proposed method.
>
> Reference
>
> [1] Song L, et al. Adversarial discriminative heterogeneous face recognition. AAAI 2018.
>
> [2] Fu C, et al. Dvg-face: Dual variational generation for heterogeneous face recognition. TPAMI 2021.
>
> [3] Lattas A, et al. AvatarMe++: Facial shape and BRDF inference with photorealistic rendering-aware GANs. TPAMI 2021.

---

### Official Review · Reviewer_Mb2x · 2022-07-13

**Rating:** 5
**Confidence:** 5
**Soundness:** 3 good
**Presentation:** 3 good
**Contribution:** 3 good

**Summary:**

The work proposes a NIR-VIS face matching dataset constructed with a physically-based renderer. An ID-MMD loss is employed to facilitate the identity feature learning as well as reduce the modality discrepancy. The work achieves state-of-the-art performance on 4 NIR-VIS face recognition benchmarks.

**Questions:**

(1)	It would be better if the combinations of modality discrepancy reduction losses and id loss, as the combination of losses sometimes can have larger impact than single ones.
(2)	In DVG-Face, the evaluation metrics of generation quality are Mean Similarity, Mean Instance Similarity and Frechet Inception Distance. Why do the authors take different metrics in this work? Table 2 shows that the proposed method holds the smallest Mean Identity feature Distance. However, it can also be interpreted as lack of diversity.


**Ethics Review Area:**

["I don’t know"]

**Limitations:**

See Weaknesses & Questions

**Strengths And Weaknesses:**

Strengths:
(1)	The proposed method is capable to automatically generate multiple NIR-VIS image pairs with identity information reserved, which is of great significance.
(2)	With the proposed training scheme, the NIR-VIS face matching dataset helps improve the NIR-VIS face recognition performance by a large margin.

Weaknesses:
(1)	The proposed ID-MMD loss reduces the distance between the NIR-VIS feature centroids of the same identity, which is effective yet not novel [1].
[1] Wei, Ziyu, et al. "Syncretic modality collaborative learning for visible infrared person re-identification." ICCV. 2021.
(2)	The training differences from other compared methods should be more detailly stated
(3).  Some related works are missing, e.g., Dual face alignment learning network for NIR-VIS face recognition, Orthogonal modality disentanglement and representation alignment network for NIR-VIS face recognition, Dual-Agent GANs for Photorealistic and Identity Preserving Profile Face Synthesis

---

> ### Author Response · Authors · 2022-08-02
> **Response to Reviewer Mb2x**
>
> The suggestions are helpful, and we are open to further discussions.
> * Q1: The ID-MMD loss is not novel [1].
>
> SMCL [1] uses a tri-directional center-based loss ($L_{tricenter}$) to handle the distance between the syncretic modality and VIS/NIR modalities. Although we both focus on the relationship between the feature centroids, our ID-MMD loss differs from SMCL in:
> * SMCL regularizes the feature relationship in Euclidean space while ours is in Reproducing Kernel Hilbert Space. When linear kernels are adopted, ours is degenerated to a simple version of SMCL, i.e., only positive centroid pairs are involved.
> * Compared to SMCL, ours excludes the involvement of an intermediary modality.
>
> To illustrate the differences, we replace $L_{idmmd}$ with $L_{tricenter}$ when training on LAMP-HQ. LC-29$^\dagger$+Fake($L_{id}$) in the paper is adopted as the backbone model (B). We have following results,
>
> Model|FAR=0.01%|Rank-1
>  :-: | :-: | :-:
> B | 84.9$\pm$1.6|98.4$\pm$0.3
> B+$L_{tricenter}$| 90.5$\pm$1.5|98.8$\pm$0.3
> B+$L_{idmmd}$|**92.0$\pm$1.5**|**98.9$\pm$0.3**
>
> As can be seen, $L_{tricenter}$ is inferior to $L_{idmmd}$.
> * Q2: Training differences.
>
> The generation of NIR-VIS images and the training of NIR-VIS face recognition network do not require any existing NIR-VIS face recognition datasets.
> * Q3: Missing works [2-4].
>
> DA-GAN [4] reveals that high-quality profile view synthesis could facilitate the face recognition task. But DA-GAN is proposed for the VIS face recognition task while ours is for NIR-VIS face recognition. DFAL [3] and OMDRA [2] focus on domain-invariant face features extraction. Both methods do not involve any facial image generation with new identities.
>
> Discussion about [2-4] have been added to Background and Related Work (Section 2) and performance comparisons with [2-3] have been added to Table 5 in the revision.
> * Q4: Combinations losses.
>
> As stated in Eq. (7) and Section 4.2 (Line 255) in the paper, we employ the combination of modality discrepancy reduction losses and id loss during training. Model performances in Table 4 prove "the combination is better than single ones".
> * Q5: Different metrics with DVG-Face.
>
> We did not take the same metrics as DVG-Face due to the differences in the generation method and the training process.
> * Even though DVG-Face can generate multiple pairs of NIR-VIS images for a particular identity, it only generates one NIR-VIS pair per person. DVG-Face measures Mean Similarity (MS) between the pair to evaluate intra-identity consistency. However, we generate multiple NIR and VIS face images for a given identity. To obtain the intra-identity consistency, the feature distances (similarity) across multiple images are calculated, namely Mean Identity feature Distance (MID) in our work. In the revision, for better understanding, we compare with DVG-Face on LAMP-HQ in terms of MS between pairs and MS across multiple images, which are indicated by 1v1 and 1vN, respectively. The results have been added to Table 2 in the revision. The results show that our method outperforms DVG-Face by achieving higher MS on both settings, which proves our generation well preserves intra-identity consistency. Additionally, the 1vN MS of our method is 0.411. Given the general identity verification threshold (around 0.3), our generation preserves the faces diversity.
> * DVG-Face obtains identity representations for the face generation via random noise sampling. The evaluation of inter-identity diversity via Mean Instance Similarity (MIS) proves the low overlap between generated identities. However, the identity features we used for the face generation come from a benchmark VIS face recognition dataset (CelebA). There is no overlap between identities. Thus, we did not evaluate MIS in our work. In the revision, we add the comparison results on MIS in Table 2. Following the settings in DVG-Face, the comparisons are conducted between VIS-VIS pairs and NIR-VIS pairs. The results suggest that our generation achieves a higher inter-identity diversity than DVG-Face.
> * Frechet Inception Distance (FID) is widely used in GAN-based generation, but we use physical rendering based generation. Following DVG-Face, we also employed LightCNN for FID evaluation in the revision. The proposed method exhibits higher feature distribution
> consistency with real data than the GAN-based DVG-Face. Even though our method has not rendered
> hair and torso, our generation is more close to the feature of real data from the view of a face recognition network.
>
> Reference:
>
> [1] Wei Z, et al. Syncretic modality collaborative learning for visible infrared person re-identification. ICCV 2021.
>
> [2] Hu W, et al. Orthogonal modality disentanglement and representation alignment network for NIR-VIS face recognition. TCSVT 2021.
>
> [3] Hu W, et al. Dual face alignment learning network for NIR-VIS face recognition. TCSVT 2021.
>
> [4] Zhao J, et al. Dual-agent gans for photorealistic and identity preserving profile face synthesis. NIPS 2017.

---

### Official Review · Reviewer_Srya · 2022-07-14

**Rating:** 7
**Confidence:** 3
**Soundness:** 3 good
**Presentation:** 2 fair
**Contribution:** 3 good

**Summary:**

This paper proposed a new face rendering technique to synthesize paired NIR-VIS images for improved face recognition in NIR space. Unlike the previous methods, which use image-to-image translation models learned from paired NIR-VIS images, this method use physical-based 3D face rendering. It first reconstructs 3D face meshes and reflectance assets in VIS space, then infers the corresponding reflectance assets in NIR space, and finally synthesizes paired VIS-NIR images at various head poses and illuminations. It also employs a novel ID-MMD loss to close the gap between VIS and NIR features in NIR-VIS face recognition training. The proposed method helped to achieve state-of-the-art face recognition performance on four NIR-VIS benchmarks.

**Questions:**

- Can we improve the method by using full-head 3D reconstruction, augmented expressions, and augmented accessories?
- There is a predefined subset of WebFace260M called WebFace4M. The authors said they randomly selected images to create their WebFace4M dataset. Are they the same dataset? If not, better change the dataset name to avoid confusion.

**Limitations:**

There is no discussion on limitations. The paper limits its application to avoid potential social impacts.

**Strengths And Weaknesses:**

### Strengths
- The proposed method allows generating infinite pairs of VIS-NIR facial images covering different head poses and illuminations without learning from any real VIS-NIR image pairs.
- The paper proposes a sophisticated algorithm to transform a VIS reflectance asset into its NIR version.
- The paper proposes a novel ID-MMD loss to close the gap between VIS and NIR features in NIR-VIS face recognition training. Ablation studies confirm its effectiveness.
- By using only the synthesized VIS-NIR images, the trained face recognizers can produce competitive NIR face recognition performance compared with the baseline methods trained on real NIR-VIS datasets. After fine-tuning on real NIR-VIS images, these recognizers provide state-of-the-art and near-perfect performance on four NIR-VIS face recognition benchmarks.

### Weaknesses
- The synthesized images, particularly the VIS ones, look pretty unrealistic. The 3D models only cover the facial part, leaving some components missing, including hair, facial accessories, and background.
- The denotations such as \mathcal{R}^{NIR} should be explained earlier, e.g., in the text or the caption of Table 1, rather than in the caption of Fig. 4.
- There is a predefined subset of WebFace260M called WebFace4M. The authors said they randomly selected images to create their WebFace4M dataset. Are they the same dataset? If not, better change the dataset name to avoid confusion.
- It seems the authors did not augment facial expressions; all qualitative figures of synthetic images have neutral expressions. I think the expression is not an important factor in the test benchmarks, but the authors should consider it when developing face recognizers for real-world applications.

---

> ### Author Response · Authors · 2022-08-02
> **Response to Reviewer Srya**
>
> We thank Reviewer Srya for the feedback and suggestions. The suggestions are helpful, and we are open to further discussions.
>
> * Q1: "The synthesized images, particularly the VIS ones, look pretty unrealistic. The 3D models only cover the facial part, leaving some components missing, including hair, facial accessories, and background."
>
> There are multiple reasons for not adopting a full-head 3D model, with hair and accessories. 1\) First is the input pre-processing configuration [1] of the face recognition networks. Specifically, before training, face images are cropped according to 5 facial points (two eyes, nose and two mouth corners), where only the main facial components are included. In [2], most of discriminative facial regions are around eyes, noses and mouths. 2) Apart from the generated images, we also use a benchmark VIS face recognition dataset (WebFace4M) [3] for training, which is collected from real-world scenarios with diverse hair styles, accessories, and backgrounds and thus ensures diversity and authenticity in training data samples. Given both the aforementioned points, hair and accessories are not required to be augmented during the NIR-VIS facial image generation. In terms of background, as we can see in Figure 1 and 2 in the main paper, we do vary the background in the rendered images.
>
> Moreover, despite recent approaches in photorealistic human head and body rendering, reconstructing the human face in conjunction with the hair, torso and accessories still remains an open problem. The most relevant work of human VIS-rendering [4] utilizes only a head geometry PCA model and requires manually created rendering assets (hair, facial hair, clothes) that require professional desingers to be procured. Furthermore, photorealistic hair rendering is much slower, given the huge amount of vertices required,
> and requires complex commercial rendering engines, which cannot be easily engineered to render in NIR.
>
> In the end, a main finding of our work is that facial NIR-VIS pairs of rendered facial images can augment the capabilities of the recognition network, even without including the hair and torso.
>
> * Q2: "The denotations such as $\mathcal{R}^{NIR}$ should be explained earlier."
>
> We introduced the concept of VIS (NIR) Renderer $\mathcal{R}^{VIS(NIR)}$ in Section 3.1 (Line 167) in the main paper, i.e., "We define a VIS physically-based rendering function $\mathcal{R}^{VIS}$ …". We apologize for the misunderstanding caused by the inconsistency terms, i.e., "rendering function" and "renderer". We have clarified this term in the revision (Line 172 and 175).
>
> * Q3: "Are WebFace260M and WebFace4M the same dataset?"
>
> In [3], WebFace260M is randomly divided into 10 folds, and the first fold serves as WebFace4M. We also use this fold and it is the same subset as in [3]. We have added explanations to avoid confusions in the revision (Line 256-258).
>
> * Q4: "No facial expressions augmentations; all qualitative figures of synthetic images have neutral expressions. The authors should consider it when developing face recognizers for real-world applications."
>
> We conduct a comparison on the LAMP-HQ dataset to validate the effectiveness of facial expressions augmentations. NIR-VIS facial images with diverse facial expressions are generated by using blend-shapes. The comparison results of models trained with generated data without (w/o) and with (w/) Expressions (E) augmentations are illustrated as follows.
>
> | Setting | FAR=0.01\% | FAR=0.1\% | Rank-1 |
> | :-----:| :-----:| :----: | :----: |
> | w/o E | 91.75$\pm$1.5 | 97.96$\pm$0.3 |98.87$\pm$0.3 |
> | w/ E |  **92.05$\pm$1.5** |  **98.02$\pm$0.3** |  **98.91$\pm$0.3** |
>
> Seen from the results, it is clear that performance improvements brought by the expression augmentations are subtle, i.e., less than 0.1$\%$ in terms of Rank-1 accuracy.
>
> References
>
> [1] Wang H, et al. Cosface: Large margin cosine loss for deep face recognition. CVPR 2018.
>
> [2] Wang Q, et al. Hierarchical pyramid diverse attention networks for face recognition. CVPR 2020.
>
> [3] Zhu Z, et al. WebFace260M: A Benchmark for Million-Scale Deep Face Recognition. TPAMI 2022.
>
> [4] Wood E, et al. Fake it till you make it: face analysis in the wild using synthetic data alone. ICCV 2021.

---

> > ### Comment · Reviewer_Srya · 2022-08-06
> > **Thanks for your answers**
> >
> > Thanks for your answers. They addressed my concerns pre-rebuttal.
> > I decide to keep my Accept score.

---

### Author Response · Authors · 2022-08-02
**Response to All Reviewers**

We sincerely thank all reviewers for their valuable comments and insightful advice on our paper. We are pleased to see that all reviewers give highly positive ratings (one accept and two borderline accepts). The main changes are highlighted in blue in the revision and our response to all comments can be found as follows.

---

### Meta-Review · Area_Chair_J8pg · 2022-08-25

**Recommendation:** Accept
**Confidence:** Certain

**Metareview:**

The paper received 3 positive reviews. The reviewers all lean towards acceptance after the rebuttal.

Overall this work can be of large interest to the community working on NIR-VIS Recognition. But I hope the authors will present additional visualized results, as suggested by the reviewers.

**Award:**

No

---

### Decision · Program_Chairs · 2022-09-14

Accept